# Hexagonal Boron Nitride Passivation Layer for Improving the Performance and Reliability of InGaN/GaN Light-Emitting Diodes

**Gun-Hee Lee [1], Tran-Viet Cuong [2], Dong-Kyu Yeo [1], Hyunjin Cho [3], Beo-Deul Ryu [1], Eun-Mi Kim [4], Tae-Sik Nam [4], Eun-Kyung Suh [1], Tae-Hoon Seo [4],* and Chang-Hee Hong [1],***

[1] School of Semiconductor and Chemical Engineering, Semiconductor Physics Research Center, Chonbuk National University, Jeonju 54896, Korea; leegunhe7@jbnu.ac.kr (G.-H.L.); dkyeo@jbnu.ac.kr (D.-K.Y.); lbd0906@jbnu.ac.kr (B.-D.R.); eksuh@jbnu.ac.kr (E.-K.S.)

[2] VKTech Research Center, NTT Hi-Tech Institute, Nguyen Tat Thanh University, 298-300A Nguyen Tat Thanh Street, District 4, Ho Chi Minh City 70000, Vietnam; tvcuong@ntt.edu.vn

[3] Institute of Advanced Composite Materials, Korea Institute of Science and Technology, Jeobuk 55324, Korea; hyunjincho34@gmail.com

[4] Green Energy & Nano Technology R&D Group, Korea Institute of Industrial Technology, Gwangju 61012, Korea; kimeunmi@kitech.re.kr (E.-M.K.); tsnam@kitech.re.kr (T.-S.N.)

\* Correspondence: thseo@kitech.re.kr (T.-H.S.); chhong@jbnu.ac.kr (C.-H.H.)

**Abstract:** We introduce a low temperature process for coating InGaN/GaN light-emitting diodes (LEDs) with h-BN as a passivation layer. The effect of h-BN on device performance and reliability is investigated. At $-5$ V, the leakage current of the h-BN passivated LED was $-1.15 \times 10^{-9}$ A, which was one order lower than the reference LED's leakage current of $-1.09 \times 10^{-8}$ A. The h-BN layer minimizes the leakage current characteristics and operating temperature by acting as a passivation and heat dispersion layer. With a reduced working temperature of 33 from 45 °C, the LED lifetime was extended 2.5 times following h-BN passivation. According to our findings, h-BN passivation significantly improves LED reliability.

**Keywords:** light emitting diodes; h-BN; passivation; operating temperature

## 1. Introduction

Due to their low power consumption, great brightness and long lifetime, GaN-based light-emitting diodes (LEDs) have been employed in solid state lighting applications such as lamps, headlights, plant growth, and lighting communication [1]. Although various research has offered ways to improve the efficiency of light-emitting devices, lifetime degradation owing to heat generation and leakage current remains an issue [2–6]. Due to the different resistances of the electrode and devices, heat is generated around the electrode contact in the LED current injection, and the device's operational temperature steadily rises, resulting in a shorter lifetime. Passivation is one of the methods for reducing leakage current and increasing reliability, and insulator materials such as $AlN_x$, $SiN_x$, and $SiO_2$ are commonly utilized [7,8]. However, passivation of these insulating materials necessitates a plasma-enhanced chemical vapor deposition (PECVD) approach with a high temperature process. This produces poor p-type ohmic contact between metal and LED surfaces, which might result in device performance impairment during the PECVD process [6]. New technologies or materials with little effect on the device surfaces or contacts are required to achieve high LED reliability.

Recently, researchers used hexagonal boron nitride (h-BN) as a passivation layer to increase the performance of high electron mobility transistors and solar cells [9–11]. One of the most notable advantages of using h-BN as a passive layer over other insulators is that the process of applying it to the device does not require high temperatures. Moreover,

h-BN, sometimes known as white graphene, is a two-dimensional material with a structure similar to graphene, a III–V compound containing covalently bound boron and nitrogen atoms. High flexibility, high transmittance, high strength, high thermal conductivity, high temperature stability, and insulator qualities are all characteristics of an h-BN thin film [12–15].

In this study, we introduce h-BN as a passivation layer to improve the reliability of GaN-based LEDs. For material properties, scanning electron microscopy (SEM), transmission electron microscopy (TEM), and Raman spectroscopy were used to examine the produced h-BN film. The performance of the LEDs with h-BN passivation was compared to that of normal LEDs by measuring leakage current, electroluminescence, and light output power. Temperature measurement using an infrared camera validated the heat dispersion effect of the h-BN layer.

## 2. Materials and Methods

### 2.1. Device Fabrication

Metal organic chemical vapor deposition was employed to create the GaN-based LEDs in this study on a c-plane sapphire substrate. The un-doped GaN layer was grown to a thickness of 1.5 μm at 1040 °C after a GaN buffer layer was grown to a thickness of 25 nm at 550 °C. A 2 μm n-type GaN layer was then grown at 1100 °C using silane ($SiH_4$) as a doping source. Following that, 5 pairs of $In_{0.04}Ga_{0.96}N$/GaN multi-quantum wells and a p-type GaN were grown on a substrate using $NH_3$, trimethylindium, triethylgallium, and $Cp_2Mg$ to create a GaN-based LED structure. The Mg dopant of p-type GaN was activated by rapid thermal annealing at 940 °C in a $N_2$ environment. To fabricate the LEDs, a photo-lithography procedure and reactive ion etching with $BCl_3$/$Cl_2$ gases were used to implement the mesa pattern structure to obtain an n-type GaN contact layer. Finally, a transparent and conductive electrode of 200 nm of indium tin oxide was formed on the selected region above the p-type GaN using electron beam evaporator, and a Cr and Au bimetal stack was deposited to 50 and 200 nm, respectively, to form n-type and p-type pads for current injection into LEDs.

To grow the h-BN film on the polycrystalline Cu foil, 0.3 sccm borazine ($B_3N_3H_6$) and 70 sccm $H_2$ were introduced in a thermal chemical vapor deposition chamber at 1100 °C for 5 min at $7 \times 10^{-3}$ torr. The Cu foil was etched with Cu etchant to form a PMMA/h-BN structure after the h-BN/Cu foil was covered with PMMA (polymethylmethacrylate), using a spin-coater at 4000 rpm for 1 min. The PMMA/h-BN structure was rinsed with de-ionized water to remove residual Cu etchant. Finally, the PMMA/h-BN structure was transferred onto the fabricated LEDs, and the PMMA was removed by acetone treatment for 1 h. After h-BN transfer onto LEDs, the h-BN on the n-type and p-type pads was selectively removed using a photo-lithography process, and reactive ion etching with $O_2$ plasma for 10 s was used for measuring the electrical and optical characteristics of LEDs. Figure 1 shows the h-BN-passivated GaN-based LED structure.

### 2.2. Device Characterization

A field emission scanning electron microscope (Nova Nano SEM 450) and Raman spectroscope with an argon ion laser (514 nm) were used to access the crystalline quality of the produced h-BN layer. The electrical performance of h-BN-passivated GaN-based LEDs was measured using a semiconductor characterization system (Keithely 4200-MSTech). An optical detector linked to a parameter analyzer was used to quantify electroluminescence (EL) at a current injection of 20 mA and optical output power was measured as a function of applied current (L-I). Thermal imaging was carried out using a middle wavelength infrared detector and an active thermography system instrument.

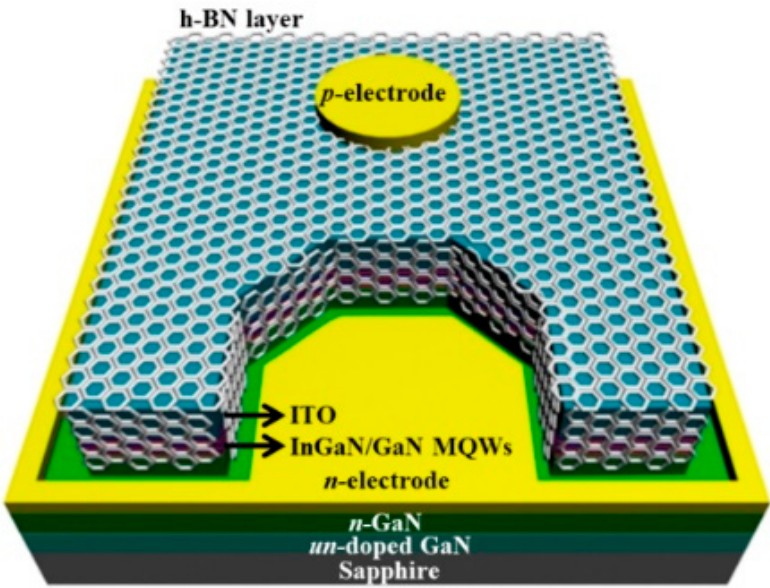

**Figure 1.** Schematic diagram of fabricated InGaN/GaN-based light-emitting diode (LED) with hexagonal boron nitride (h-BN) film as a passivation layer.

## 3. Results and Discussion

To analyze the h-BN layer itself, the produced h-BN on polycrystalline Cu foil was transferred to a Si substrate. The scanning electron microscopy (SEM) images of the h-BN layer after it was relocated to a Si substrate is shown in Figure 2a,b. These indicate that the h-BN layer has a tearing-free two-dimensional nanosheet structure. However, there are a few wrinkles on the surface. The discrepancy in thermal expansion coefficients between Cu foil and h-BN was blamed [16]. A transmission electron microscopy image of the atomic arrangement of the h-BN sheet is shown in Figure 2c. The organization of B and N atoms in a single hexagonal lattice was clearly observed [17]. Furthermore, because Raman spectroscopy has become an important method for characterizing and investigating two-dimensional materials, we used it to further study the h-BN monolayer crystal structure. Figure 2d shows the Raman spectra of transferred h-BN on a Si substrate; it showed a G band centered at 1370 cm$^{-1}$, which corresponds to the $E_{2g}$ vibration mode in h-BN. It was reported that the G band frequency of bulk h-BN was observed to be at $1366.6 \pm 0.2$ cm$^{-1}$, which upshifted to $1369.6 \pm 0.6$ cm$^{-1}$ when the substrate was atomically monolayer h-BN. This makes sense when the stronger $E_{2g}$ phonons are considered, which result in a blueshift in the G band. In addition, the full width at half maximum was only 21.7 cm$^{-1}$. These signatures indicating high crystalline quality of h-BN monolayers can be verified using both TEM and Raman techniques [18,19].

Figure 3a,b show the current-voltage (I-V) characteristics of manufactured LEDs with and without an h-BN passivation layer at forward and reverse bias, respectively. The forward voltages for both devices are 3.8 V at a 20 mA injection current. Because the h-BN layer has no influence on the device, contact resistance, or stress, the results are the same for both devices in forward bias. Figure 3b shows that the leakage current drops by about one order from $1.09 \times 10^{-8}$ to $1.15 \times 10^{-9}$ A at $-5$ V, when the bias is reversed. This finding suggests that the monolayer h-BN is effective in LED passivation, lowering the reverse leakage current through the LED's surface. The leakage current is minimized because the h-BN successfully passivates the defects on the device's surface and mesa sidewall [20]. The EL at 20 mA and light output power of both devices are shown in Figure 3c,d, respectively. Both measurement results show that after h-BN passivation, light output performance is somewhat improved. In the visible range of 460 nm, which is the wavelength of the fabricated LEDs, monolayer h-BN has a transmittance of more than 90%, which means that the majority of the light is transmitted. The device's leakage current was lowered after h-BN application due to a passivation effect, and the intensity of EL was improved. In

comparison to the traditional LED, the EL peak of the LED with h-BN exhibits a minor blue shift. The blue shift in the LED with h-BN passivation can be attributed to a reduced band gap shrinking due to efficient heat dispersion via h-BN [21].

Figure 4a,b are infrared camera images during the operation of reference LEDs and LEDs with h-BN passivation, respectively. As seen in the images, heat surrounding the contact is not concentrated, as shown by an infrared camera image of a 350 μm × 350 μm area LED with h-BN passivation. Most of the heat is generated by current crowding in the p-type contact, which causes the temperature of the device to rise [22]. Temperature measurement was performed at 100 mA current injection to confirm the change in operation temperature after h-BN passivation; the average temperature for the LED with h-BN passivation was 33 °C, which is significantly lower than the temperature of 45 °C for the reference LED, as shown in Figure 4c. Both devices have a 3-min transition time from room temperature to working temperature when using a 100 mA injection current, before becoming saturated. The temperature drops because the heat is not localized just to the contact pad region during current injection and is effectively dispersed, since the thermal conductivity of h-BN (550 W/mK for basal plane, 30 W/mK for perpendicular) is substantially higher than that of GaN (253 W/mK) [23,24]. Figure 4d demonstrates how, due to an overload of 8 V, the light output drops with time. Rather than waiting for the LED to entirely fail, this overload determines the LED's expected lifetime when the light output exceeds 70% of its previous luminous intensity. Because of the operational temperature of the devices, the light output of both devices decreases over time when the overload is applied. It took roughly 1010 h for a reference LED to attain 70% of its output, which is the predicted device lifetime, and it climbed to about 2500 h following h-BN passivation. The heat dispersion effect of h-BN is responsible for the increased lifetime expectancy, which can be proved by the fact that the metal has been effectively passivated to the LEDs.

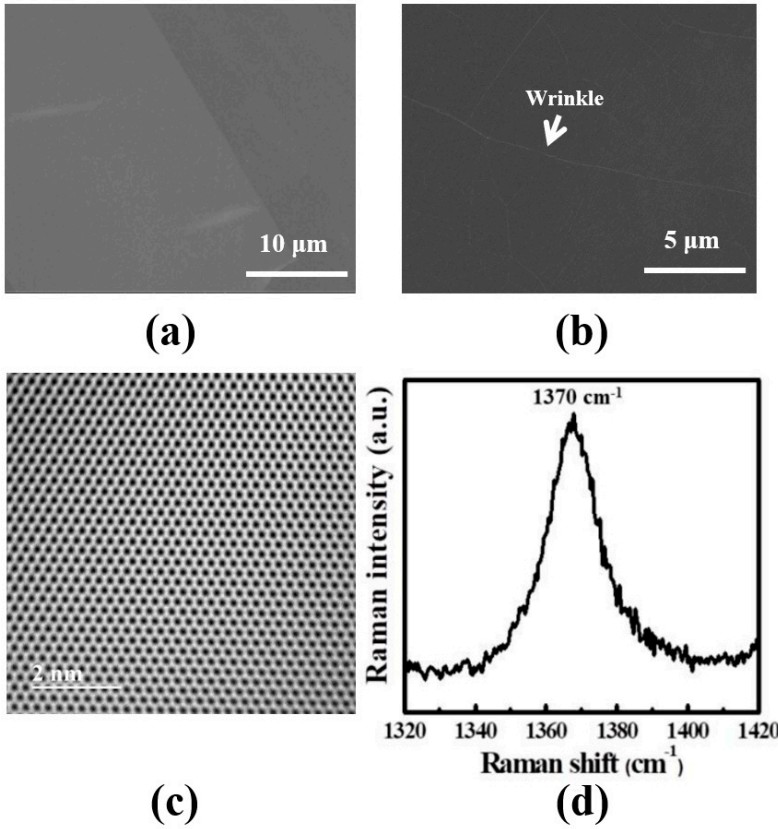

**Figure 2.** (**a**,**b**) Scanning electron microscope (SEM) image, (**c**) transmission electron microscopy (TEM) image and (**d**) Raman spectrum of h-BN film.

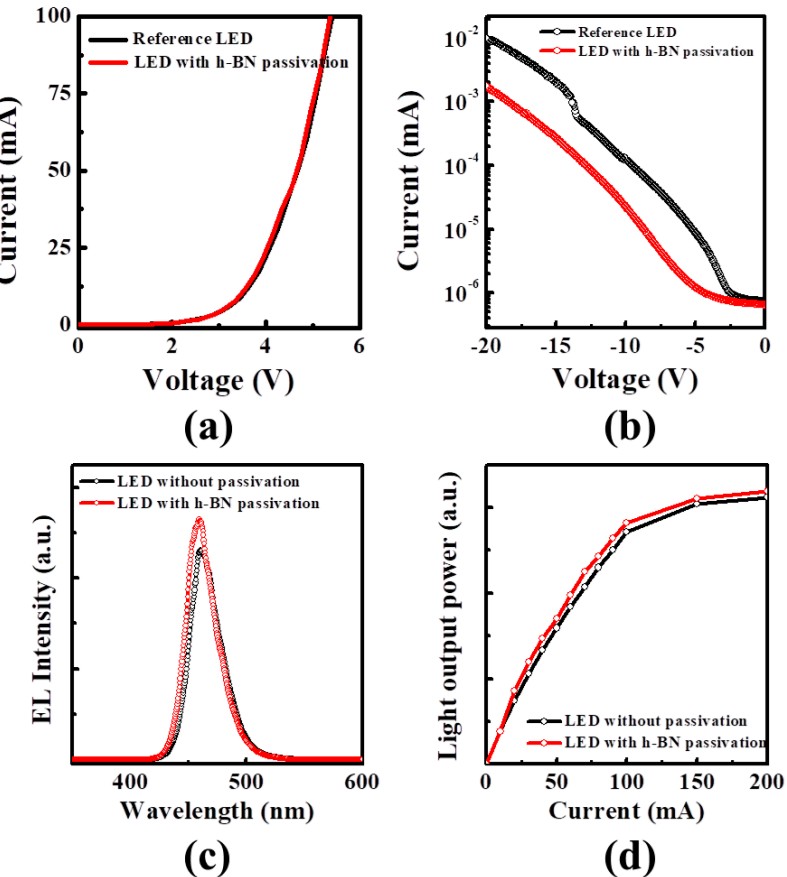

**Figure 3.** Current-voltage (I-V) characteristics of the investigated devices at (**a**) forward bias and (**b**) reverse bias. (**c**) Electroluminescence (EL) spectra and (**d**) light output power of conventional and h-BN-passivated InGaN/GaN LEDs.

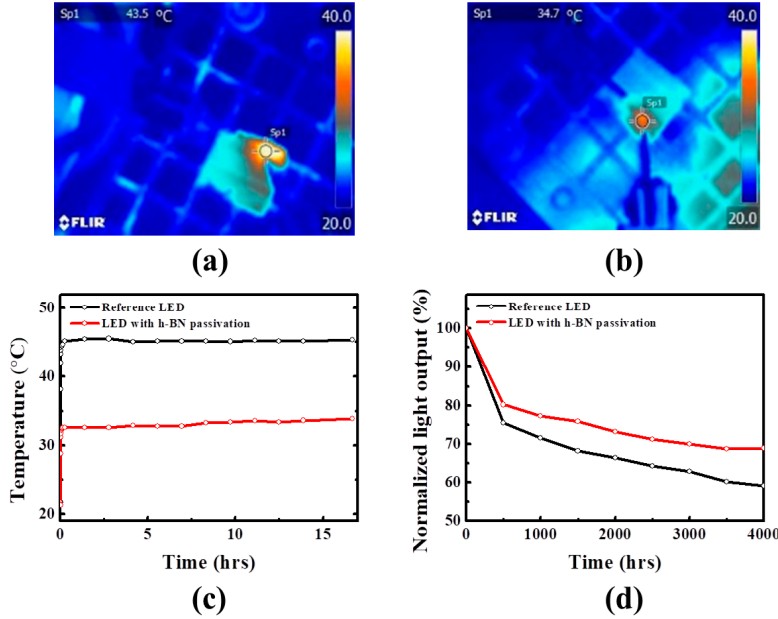

**Figure 4.** Infrared camera images during operation of (**a**) reference LED and (**b**) LED with h-BN passivation. (**c**) Operation temperatures on both devices as a function of operation time at 100 mA. (**d**) Normalized light output decay of LED with and without h-BN passivation.

## 4. Conclusions

To summarize, thermal chemical vapor deposition (TCVD) was used to synthesize high quality monolayer h-BN, which was subsequently wet transferred onto a GaN-based LED for passivation. The EL intensity and light output power of LEDs with h-BN passivation are higher than ordinary LEDs without passivation. The device's thermal stability was improved by reducing leakage current and self-heating with the help of the h-BN passivation effect. After h-BN application, the operating temperature dropped from 45 to 33 °C, and its lifetime expectancy increased by nearly 2.5 times. Because of its extraordinary thermal stability and conductivity, the monolayer h-BN can be used as a passivation and heat dispersion layer for LEDs—but it also has a lot of potential in photoelectric device applications, according to our findings.

**Author Contributions:** Conceptualization, G.-H.L., T.-H.S. and C.-H.H.; methodology, B.-D.R., E.-M.K., T.-S.N. and E.-K.S.; validation, G.-H.L., T.V.C., T.-H.S. and C.-H.H.; formal analysis, G.-H.L., D.-K.Y. and H.C.; writing—original draft preparation, G.-H.L., T.V.C. and T.-H.S.; writing—review and editing, G.-H.L., T.V.C., T.-H.S. and C.-H.H.; visualization, G.-H.L., T.V.C., D.-K.Y., H.C., B.-D.R., E.-M.K., T.-S.N., E.-K.S., T.-H.S. and C.-H.H. All authors have read and agreed to the published version of the manuscript.

**Funding:** This research received no external funding.

**Institutional Review Board Statement:** Not applicable.

**Data Availability Statement:** Data is available on request to any of the corresponding authors.

**Acknowledgments:** This research was supported by the Korea Institute of Industrial Technology (KITECH) and by the National Research Foundation of Korea (NRF) funded by the Ministry of Education, Science and Technology (2019RM3F5A02092650 and 2020R1I1A3A04036537).

**Conflicts of Interest:** The authors declare no conflict of interest.

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
