# Peer review of "Hexagonal Boron Nitride Passivation Layer for Improving the Performance and Reliability of InGaN/GaN Light-Emitting Diodes"

_applsci, doi:10.3390/app11199321_

Round 1
Reviewer 1 Report
The authors used hBN as a passivation layer for the InGaN/GaN LEDs. They observed a lower leakage current than the LEDs using other passivation methods. The passivation of LEDs is important for the stability and reliability of LEDs. Although hBN has already used for a passivation layer for many devices, this work is still a important contribution to this community. My comments are listed as below:
- hBN is a well-know passivation layer. However, the author only listed one reference in the manuscript which used hBN for the passivation layer of transistors. The relevant reference using hBN as the passivation layer of LEDs are missing.
- The authors should specify the thickness of hBN in the manuscript. According to line 107, the hBN on Cu foil is monolayer. But the thickness of hBN that directly grown on LED is unknown. The authors claimed it is monolayer hBN on LED on line 131. Since the substrate is different, more discussion or data are needed to justify the thickness of hBN.
Reviewer 2 Report
- In section 2. Materials and Methods 2.1. Device fabrication, the author should describe the material preparation process parameters and device fabrication process separately and in detail.
- Figure 1. Schematic diagram of fabricated InGaN/GaN based light emitting diode (LED) with 95 hexagonal boron nitride (h-BN) film as a passivation layer. The author should clearly indicate various functional layers in the schematic diagram.
- The scanning electron microscopy (SEM) image of h-BN 99 layer after it was relocated to a Si substrate is shown in Figure 2(a). I think it is necessary to show a wide range of micrographs here. And the h-BN layer has a tearing-free two-dimensional nanosheet structure as well as a few wrinkles on the surface.
- Figure 2(c) shows the Raman spectra of transferred h-BN on Si substrate. It showed a G band centered at 1370 cm-1 which is corresponds to the E2g vibration mode in h-BN. It was reported that the G band frequency of bulk h-BN was observed to be at 1366.6±0.2 cm-1, which upshifted to 1369.6±0.6 cm-1 when the substrate was atomically monolayer h-BN. The wave number range of Raman spectrum is too narrow to explain the atomically monolayer h-BN.
- Figure 4. (a) Operation temperatures on both devices as function of operation time at 100 mA. The inset of Figure 4 (a) heat surrounding the contact are difficult to distinguish.
